# ME-LORA: MEMORY-EFFICIENT BAYESIAN LOW-RANK ADAPTATION FOR LARGE LANGUAGE MODELS

## ABSTRACT

Bayesian Low-Rank Adaptation (LoRA) has shown excellent performance in reducing the overconfidence of inference by large language models as it can accurately quantify the inference uncertainty. However, the general Bayesian LoRA technique requires huge memory as it fine-tunes three low-rank matrices with large size: two matrices have size of $n \times r$ and the other has size of $r \times m$, where $r$ denotes rank, and $n, m$ denote the size of input and output, respectively. The large amount of memory required by this technique precludes its practical applications especially for the cases with long input or output. Here, we propose a memory efficient Bayesian LoRA technique (called Me-LoRA) that needs only two low-rank matrices plus two small matrices with size of only $r \times r$. The key idea of our approach is that we introduce a small matrix (with size $r \times r$) to describe the variance estimates required by Bayesian LoRA, which is calculated through sampling two other samll matrices. Compared with the general Bayesian LoRA technique, our approach reduces the memory requirement by nearly $\frac{1}{3}$ as the rank $r$ is generally very small. Experimental results using both LlaMA-7B and LlaMA-13B models on representative data sets suggest that our approach achieves the same performance as the original Bayesian LoRA techniques and outperforms the existing approaches. In summary, the memory-efficient Bayesian LoRA presented in this study circumvents the challenge of high memory requirement and thus paves a new way to the practical applications of Bayesian LoRA in the cases with larger input and output size.

## 1 INTRODUCTION

In recent years, efficient parameter training has become increasingly important, particularly in large-scale neural networks Hu et al. (2021); Ding et al. (2023); Fu et al. (2023). As models grow, managing and optimizing parameters is crucial for training speed and performance. This requirement has led to growing interest in techniques that can achieve comparable results with fewer trainable parameters, especially in large language models (LLMs). Parameter-efficient fine-tuning (PEFT) methods have garnered significant attention for their ability to reduce computational cost and memory usage while maintaining model performance Liu et al. (2022); Han et al. (2024).

As fine-tuning large language models (LLMs) becomes increasingly crucial, various techniques have emerged to optimize this process Houlsby et al. (2019); Hu et al. (2021); Ding et al. (2023). In fine-tuning large models, Bayesian methods are frequently employed to mitigate overconfidence, enhance factual accuracy, and reduce potential harmful consequences Amodei et al. (2016); Weidinger et al. (2021); Azaria & Mitchell (2023). Bayesian fine-tuning for large models involves incorporating probabilistic frameworks to model uncertainty in the model parameters, enabling the model to make more robust predictions in the face of uncertainty. This approach allows for considering parameter distributions rather than solely on point estimates, thus providing more reliable inferences when encountering new data.

Despite the potential advantages of Bayesian methods, current state-of-the-art (SOTA) Bayesian approaches still need to address the issue of parameter efficacy. This challenge underscores the difficulty of effectively estimating and updating parameters during complex fine-tuning tasks. Consequently, further research to enhance the performance and reliability of Bayesian fine-tuning methods is crucial for advancing the deployment of large models in practical applications.

Among the approaches to Bayesian fine-tuning, some of the most common are post-training methods like Monte Carlo Dropout (MCD) Gal & Ghahramani (2016) and Deep Ensembles (ENS) Lakshminarayanan et al. (2017); Wang et al. (2023); Balabanov & Linander (2024) only gains in generalization and uncertainty estimation remain marginal. More promising post-training approaches, such as the Kronecker-factored Laplace approximation, apply after any optimization algorithm's maximum a posteriori (MAP) estimate MacKay (1992). However, this two-step procedure inherently leads to suboptimal posterior estimates due to the separation of training and uncertainty estimation phases.

In contrast, Bayesian fine-tuning methods like BLoB Wang et al. (2024) jointly estimate the mean and covariance of a low-rank variational distribution. Unlike post-training approaches, BLoB estimates the parameter mode (i.e., the mean if one assumes Gaussian distributions) and the parameter variance simultaneously. While BLoB enhances the mode estimate by randomly sampling from the variance estimates, the method requires significant additional memory, increasing the number of trainable parameters by nearly half. BLoB introduces a storage burden that scales with the size of LLMs. Our approach improves upon BLoB by reducing the number of trainable parameters and addressing the memory overhead while retaining the advantages of Bayesian fine-tuning.

Our method, Memory-efficient Bayesian Low-Rank Adaptation (Me-LoRA), extends the benefits of BLoB, significantly optimizing the parameter efficacy. Me-LoRA reduces training parameters by approximately one-third compared to BLoB, making Bayesian uncertainty estimation feasible for larger models while retaining the accuracy and uncertainty benefits that have been proven valuable in both in-distribution and out-of-distribution settings.

In summary, this work presents a pioneering approach to optimizing Bayesian fine-tuning methods, highlighting the necessity and potential of reducing training parameters for large-scale models. Our contributions are as follows:

- We propose Memory-efficient Bayesian Low-Rank Adaptation (Me-LoRA). This novel method reduces trainable parameters by approximately one-third compared to SOTA Bayesian fine-tuning methods without compromising performance.
- Me-LoRA significantly lowers the memory overhead of Bayesian fine-tuning, making it more scalable for large language models.
- We preserve the critical advantages of Bayesian methods, including enhanced calibration and uncertainty estimation, across 7B and 13B model.
- Extensive experiments demonstrate the efficiency and effectiveness of Me-LoRA, achieving state-of-the-art performance with fewer trainable parameters.

## 2 BACKGROUND

### 2.1 LOW-RANK ADAPTATION (LoRA)

Low-Rank Adaptation (LoRA) Hu et al. (2021) is a technique that significantly reduces the number of trainable parameters in large-scale models by leveraging their inherent low-rank structure Li et al. (2018); Aghajanyan et al. (2020). In LoRA, the weight update matrix $\Delta \boldsymbol{W}$ is decomposed into two low-rank matrices:

$$\Delta \boldsymbol{W} = \boldsymbol{BA}, \tag{1}$$

where $\boldsymbol{B} \in \mathbb{R}^{m \times r}$ and $\boldsymbol{A} \in \mathbb{R}^{r \times n}$, with $r \ll \min(m, n)$. Here, $m$ and $n$ are the dimensions of the input and output layers, respectively, while $r$ represents the intrinsic rank of the decomposition. This factorization leads to a substantial reduction in the number of parameters to train, from $\mathcal{O}(mn)$ in the full-rank case to $\mathcal{O}(r(m + n))$. The forward pass of the model is then expressed as:

$$\boldsymbol{z} = \boldsymbol{W}_0 \boldsymbol{h} + \Delta \boldsymbol{W} \boldsymbol{h} = \boldsymbol{W}_0 \boldsymbol{h} + \boldsymbol{BA} \boldsymbol{h}, \tag{2}$$

where $\boldsymbol{h}$ is the input to the layer, $\boldsymbol{z}$ is the output, and $\boldsymbol{W}_0$ is the pre-trained weight matrix. By updating only the low-rank matrices $\boldsymbol{A}$ and $\boldsymbol{B}$, LoRA reduces memory consumption for storing optimizer states and accelerates fine-tuning, especially for large language models (LLMs). This method achieves performance comparable to full-rank fine-tuning, while drastically lowering the hardware requirements, making it a practical solution for fine-tuning massive models.

## 2.2 Bayesian Low-Rank Adaptation by Backpropagation (BLoB)

In Bayesian Low-Rank Adaptation by Backpropagation (BLoB) Wang et al. (2024), the posterior distribution of the model parameters is inferred rather than relying on point estimates Bishop & Nasrabadi (2006); Wang & Yeung (2020), offering a probabilistic view of the model's weights. Given the intractability of exact posterior inference, BLoB approximate the true posterior by minimizing the Kullback-Leibler (KL) divergence between the variational distribution $q(\boldsymbol{W}|\theta)$ (denoted as $q(\boldsymbol{W})$ ) and the true posterior $P(\boldsymbol{W}|\mathcal{D})$ (denoted as $P(\boldsymbol{W})$ ) Hinton & Van Camp (1993); Graves (2011); Blundell et al. (2015). We can then minimize the variational free energy concerning the variational distribution parameters $\boldsymbol{\theta}$ Neal & Hinton (1998); Yedidia et al. (2001); Friston et al. (2007):

$$\min_{\boldsymbol{\theta}} \mathcal{H}(\mathcal{D}, \boldsymbol{\theta}) = -\mathbb{E}_{q(\boldsymbol{W})}[\log P(\mathcal{D}|\boldsymbol{W})] + D_{\mathrm{KL}}[q(\boldsymbol{W})\|P(\boldsymbol{W})]. \tag{3}$$

Here, the first term corresponds to the expected log-likelihood, encouraging the model to fit the data. In contrast, the second term is a regularizer by penalizing the divergence between the variational distribution and the true posterior. This objective ensures a balance between model complexity and data fit, improving both the expressiveness and tractability of the posterior approximation. Such a formulation provides a principled framework for Bayesian learning, facilitating uncertainty estimation and regularization in neural networks.

BLoB optimizes the first term in Equation 3 using Monte Carlo gradient estimation LeCun (1985); Rumelhart et al. (1986), combined with the reparameterization trick, allowing gradients to propagate through the underlying parameters $\boldsymbol{\theta}$ Opper & Archambeau (2009); Kingma (2013); Rezende et al. (2014). The variational distribution is simplified to a diagonal Gaussian $\mathcal{N}(\boldsymbol{\mu}, \boldsymbol{\sigma}^2)$, where $\boldsymbol{\sigma} = \boldsymbol{\rho}^2$ ensures the positivity of the standard deviation and accelerates convergence. Here, $\boldsymbol{\rho}$ denote parameters of the weight matrix $\boldsymbol{W}$, expressed as $\boldsymbol{W} = \boldsymbol{\mu} + \boldsymbol{\rho}^2 \odot \boldsymbol{\epsilon}$, where $\boldsymbol{\epsilon} \sim \mathcal{N}(\boldsymbol{0}, \boldsymbol{I})$.

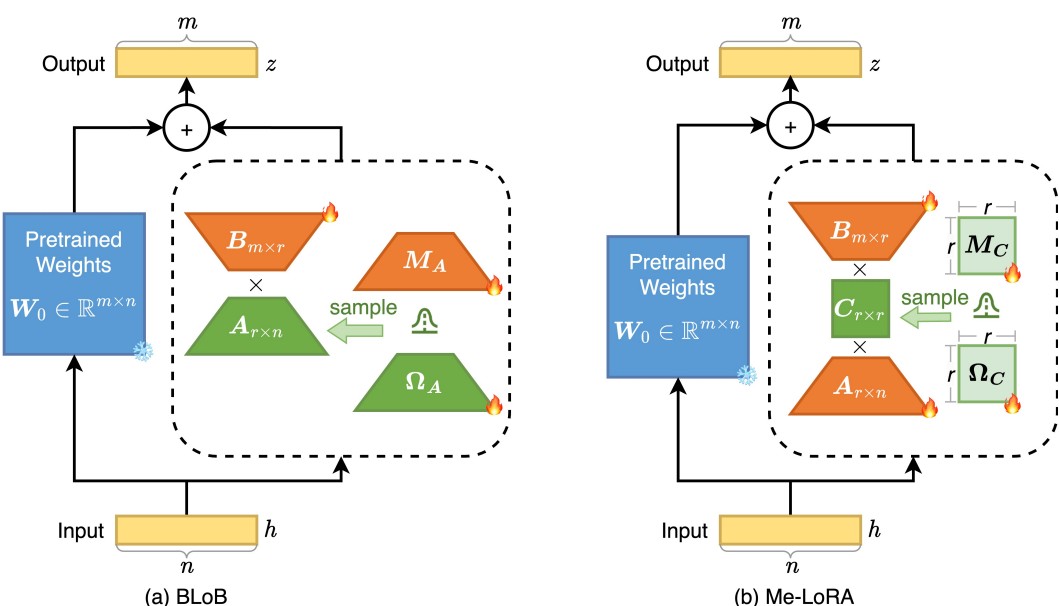

Figure 1: Overview of the general Bayesian LoRA technique BLoB (**left**) and our memory-efficient Bayesian low-rank adaptation (Me-LoRA) approach (**right**). BLoB updates the weight matrix $W_0$ using the product of two low-rank matrices $\boldsymbol{A} \in \mathbb{R}^{r \times n}$ and $\boldsymbol{B} \in \mathbb{R}^{m \times r}$, in which $r$ denotes the rank, and $\boldsymbol{A}$ is calculated through sampling two other matrices $M_A$ and $\Omega_A$ with the same size. Thus, BLoB needs three large matrices in total. In contrast, Me-LoRA introduces a small low-dimensional matrix $\boldsymbol{C} \in \mathbb{R}^{r \times r}$. This approach only requires two low rank matrices $\boldsymbol{A}$, $\boldsymbol{B}$ and two small matrices mean $\boldsymbol{M}_C$ and variance $\boldsymbol{\Omega}_C$. This way, Me-LoRA reduces the memory requirement by nearly $\frac{1}{3}$ as the rank $r$ is significantly smaller than $m$ and $n$

## 3 METHOD

In this section, we introduce Memory-efficient Bayesian Low-Rank Adaptation (Me-LoRA), a novel method that builds upon and extends the state-of-the-art method, BLoB Wang et al. (2024). The central innovation of Me-LoRA lies in the introduction of a low-dimensional matrix C, coupled with variance estimation applied to this matrix. Specifically, we introduce this low-dimensional matrix $C$ between the low-rank matrices $A$ and $B$, allowing for Bayesian modeling over the full parameter matrix, as illustrated in Figure 1. Similar to BLoB, we train the low-rank matrices $A$ and $B$, along with the mean $M$ and variance $\Omega$ of the low-dimensional matrix $C$, sampling from the mean and variance of $C$ for inference.

### 3.1 VARIATIONAL DISTRIBUTION OF THE FULL-WEIGHT MATRIX IN ME-LORA

With the pre-trained weight matrix $W_0 \in \mathbb{R}^{m \times n}$, the low-rank weight update matrix $A \in \mathbb{R}^{r \times n}$ and $B \in \mathbb{R}^{m \times r}$, we suppose that the variational distribution of the other low-rank update matrix $C \in \mathbb{R}^{r \times r}$ is Gaussian with mean $M$ and standard deviation $\Omega$, denoted as $q(C) \sim \mathcal{N}(M, \Omega)$, where $M \in \mathbb{R}^{r \times r}$ and $\Omega \in \mathbb{R}^{r \times r}$. We then have $q(Q) \sim \mathcal{N}(MA, \Omega|A|)$, where $Q = CA$. The equivalent variational distribution defined on the full weight matrix $W = W_0 + BCA$ as fallow,

$$q(W) = \mathcal{N}(W; \mu_q, \Sigma_q), \tag{4}$$

where $\mu_q = \text{vec}(W_0 + BMA)$ and $\Sigma_q = [I \otimes B] \cdot [\text{diag}(\text{vec}(\Omega|A|)^2)] \cdot [I \otimes B^\top]$.

In Equation 9 (See more details in Appendix A), our LoRA Bayesianization employs a Gaussian variational distribution for $W$ with a flexible covariance matrix $\Sigma_q$, to approximate the posterior distribution of the full parameter $W$. The covariance matrix $\Sigma_q$ is strictly singular and high-dimensional. Sampling from such a high-dimensional matrix, however, requires efficient sampling algorithms that scale with the parameter space. To ensure parameter efficiency, Blob Wang et al. (2024) introduces variance estimation in module $A$. By leveraging the multiplicative properties of matrices, this effectively estimates the entire full parameter matrix. Compared to standard LoRA Hu et al. (2021), this approach necessitates the introduction of a parameter matrix of the same size as module $A$, which increases the training parameters by 50%. To reduce the training parameters, we introduce a low-dimensional matrix $C$ between matrices $A$ and $B$ and apply variance estimation to the newly introduced matrix $C$. This modification results in nearly a 30% reduction in training parameters compared to BLoB.

### 3.2 EFFICIENT COMPUTATION OF FULL-WEIGHT KL DIVERGENCE

Direct computation of the KL Divergence between the prior and posterior distributions of $W$ is non-trivial. Direct computation of the KL Divergence between the prior and posterior distributions of $\mathbf{W}$ is non-trivial. To alleviate this, we assume the prior $P(W)$ follows a low-rank Gaussian distribution, with a mean given by the pre-trained weights $W_0$, and a covariance matrix parameterized by a rank-$rr$ matrix $R \in \mathbb{R}^{(mn) \times (rr)}$. The posterior is then given by:

$$P(W) = \mathcal{N}(W; \mu_p, \Sigma_p), \tag{5}$$

where $\mu_p = \text{vec}(W_0)$ and $\Sigma_p = RR^\top$.

Assuming that $R = [\sigma_p I \otimes B]$ and $RR^\top = BB^\top$, we can simplify the KL divergence computation by focusing on the full-weight covariance of $W$, reducing the dimensionality and parameter count. We then have:

$$D_{\text{KL}}(q(W) \| P(W)) = D_{\text{KL}}(q(Q) \| P(Q)). \tag{6}$$

Concretely, we assume that the prior distribution $P(Q)$ adheres to a low-rank structure, with each parameter in both the prior and variational distributions $q(Q) \sim \mathcal{N}(MA, \Omega|A|)$ being mutually independent. We then minimize the KL divergence term for the low-rank component, $C$ and $A$, utilizing its analytical solution as presented in Equation 3 and 6:

$$\min D_{\text{KL}}[q(Q) \| P(Q)] = \frac{1}{2\sigma_p^2}(\|MA\|_2^2 + \|\Omega|A|\|_2^2) - \sum_{ij} \log \Omega_{ij}, \tag{7}$$

the detailed derivation of Equation 7 can be found in BLoB Wang et al. (2024). In the parameterization of the Gaussian variational distribution $q(Q)$, we adopt a strategy analogous to BLoB, where the

Table 1: Theoretical memory required to store trained LoRA, BLoB and ours weights for LlaMA2-7B and LlaMA2-13B models. We applie LoRA on query and key layers of each transformer block.

| Rank | Method | Llama2-7B | | Llama2-13B | |
|---|---|---|---|---|---|
| | | # Params | Required Bytes | # Params | Required Bytes |
| 8 | LoRA | 4.19M | 15.98MB | 6.55M | 24.99MB |
| | BLoB | 6.29M | 23.99MB | 9.84M | 37.54MB |
| | Ours | 4.20M | 16.02MB | 6.56M | 25.02MB |
| 64 | LoRA | 33.55M | 127.79MB | 52.43M | 200.00MB |
| | BLoB | 50.33M | 191.99MB | 78.97M | 301.25MB |
| | Ours | 34.08M | 130.00MB | 53.08M | 202.48MB |

mean matrix $M|A|$ of $q(Q)$ is directly parameterized as the output of a neural network. To ensure that each entry of the diagonal covariance matrix $\Omega$ (i.e., the standard deviation) of $q(Q)$ remains non-negative, we employ an element-wise parameterization, $\Omega_{ij} = H_{ij}^2$, where $H = [H_{ij}] \in \mathbb{R}^{r \times r}$ is a real parameter matrix determining the standard deviation $\Omega$.

### 3.3 Designing Priors in Bayesian Models

In Bayesian neural networks, designing an appropriate prior is crucial for mitigating overfitting and improving generalization Wilson & Izmailov (2020). In BLoB, the authors adopt a relatively simple prior structure, setting all prior variances to the same value, $\sigma_p$. However, with our proposed method, using such a simplistic prior variance can still lead to overfitting. To address this, we introduce a noise term $\varepsilon$, where $\varepsilon \sim \mathcal{U}(0, 1)$, into the prior variance $\sigma_p$, resulting in $\sigma_p + \varepsilon$. The closed-form solution for the KL divergence in our approach is given as:

$$\min D_{\mathrm{KL}}[q(Q)\|P(Q)] = \frac{1}{2(\sigma_p + \varepsilon)^2} \left( \|MA\|_2^2 + \|\Omega|A|\|_2^2 \right) - \sum_{ij} \log \Omega_{ij}. \tag{8}$$

### 3.4 Parameter Count

We denote the number of fine-tuned layers as $L_{\mathrm{tuned}}$ and the dimension of these layers as $d_{\mathrm{model}}$. In Me-LoRA, the number of trainable parameters is governed by $|\Theta| = L_{\mathrm{tuned}} \times (2 \times d_{\mathrm{model}} + r) \times r$, whereas in the state-of-the-art method BLoB, it is given by $|\Theta| = 3 \times L_{\mathrm{tuned}} \times d_{\mathrm{model}} \times r$. Specifically, for the lowest rank (i.e., $r = 8$), Me-LoRA requires nearly 30% fewer trainable parameters than BLoB. Furthermore, as the rank and model size increase, the parameter count in Me-LoRA increments by $3L_{\mathrm{tuned}}d_{\mathrm{model}}$, leading to significant savings compared to LoRA's $3L_{\mathrm{tuned}}d_{\mathrm{model}}$ parameter scaling. This efficiency becomes crucial in intense models, such as GPT-3 Brown (2020), which has 96 attention layers.

Based on this efficiency, the advantage of Me-LoRA is its reduction in memory usage for storing the weight adjustments post-training while maintaining the uncertainty estimation benefits of BLoB. A comparison of memory efficiency between Me-LoRA, LoRA, and BLoB is presented in Table 1.

## 4 Results

We implemented Me-LoRA using the PEFT library Mangrulkar et al. (2022) and fine-tuned LlaMA2-7B and LlaMA2-13B models on six common-sense reasoning tasks. We applied the LoRA to the queries and values of all attention layers. For hyperparameters, we strictly adhered to the default settings in both the PEFT library and the original LoRA paper Mangrulkar et al. (2022); Hu et al. (2021) to ensure reproducibility. We used a batch size of 4 and saved model checkpoints every 100 steps, training for a total of 5000 steps. For common-sense reasoning tasks, we optimized the model to predict the most probable next token corresponding to the correct answer in each dataset. We followed the dataset split strategy as in Laplace LoRA for training and validation sets during training and validating. However, for the BoolQ dataset, which lacks an official test set, we divided

Table 2: Performance of different methods applied to LoRA on Llama2-13B pre-trained weights. The evaluation is done across six common-sense reasoning tasks with a shared hyper-parameter setting after 5,000 training steps. "↑" and "↓" indicate that higher and lower values are preferred, respectively. **Bold**: the best. Underline: the second best.

| Metric | Method | WG-S | ARC-C | ARC-E | WG-M | OBQA | BoolQ |
|--------|--------|------|-------|-------|------|------|-------|
| ACC ↑ | MAP | $72.35_{0.75}$ | $72.78_{0.46}$ | $88.06_{0.67}$ | $79.31_{0.26}$ | $83.93_{1.16}$ | $83.85_{0.63}$ |
| | MCD | $69.51_{0.04}$ | $71.72_{0.07}$ | $87.91_{0.02}$ | $77.64_{0.03}$ | $83.76_{0.14}$ | $\mathbf{84.18_{0.07}}$ |
| | ENS | $70.01_{0.91}$ | $72.21_{1.01}$ | $88.06_{0.44}$ | $78.99_{0.56}$ | $\underline{84.13_{0.41}}$ | $83.87_{0.84}$ |
| | LAP | $\mathbf{72.64_{0.16}}$ | $71.53_{0.10}$ | $\underline{88.10_{0.07}}$ | $76.45_{0.09}$ | $83.26_{0.18}$ | $83.73_{0.07}$ |
| | BLoB | $67.81_{0.05}$ | $71.61_{0.02}$ | $87.81_{0.07}$ | $75.02_{0.09}$ | $82.81_{0.22}$ | $82.79_{0.14}$ |
| | Ours | $68.67_{0.07}$ | $\mathbf{72.82_{0.08}}$ | $\mathbf{88.56_{0.03}}$ | $\mathbf{79.38_{0.03}}$ | $\mathbf{85.17_{0.03}}$ | $83.74_{0.05}$ |
| ECE ↓ | MAP | $21.99_{2.26}$ | $17.66_{1.66}$ | $9.70_{0.26}$ | $15.16_{0.61}$ | $10.96_{0.88}$ | $4.47_{0.27}$ |
| | MCD | $14.38_{0.04}$ | $7.29_{0.09}$ | $7.70_{0.04}$ | $12.85_{0.05}$ | $\mathbf{4.05_{0.35}}$ | $2.62_{0.19}$ |
| | ENS | $10.99_{0.97}$ | $6.13_{1.39}$ | $7.09_{0.74}$ | $6.84_{2.05}$ | $10.07_{0.82}$ | $2.72_{0.88}$ |
| | LAP | $18.48_{0.26}$ | $\mathbf{4.18_{0.18}}$ | $\underline{2.41_{0.07}}$ | $\mathbf{2.07_{0.13}}$ | $7.81_{0.40}$ | $\underline{2.11_{0.22}}$ |
| | BLoB | $\mathbf{5.59_{0.08}}$ | $\underline{5.46_{0.13}}$ | $4.58_{0.04}$ | $8.73_{0.16}$ | $6.23_{0.51}$ | $\mathbf{1.05_{0.19}}$ |
| | Ours | $9.25_{0.06}$ | $7.60_{0.16}$ | $\mathbf{2.60_{0.05}}$ | $\underline{4.94_{0.03}}$ | $\underline{5.38_{0.06}}$ | $3.36_{0.06}$ |
| NLL ↓ | MAP | $1.14_{0.13}$ | $1.01_{0.09}$ | $0.69_{0.06}$ | $0.73_{0.04}$ | $0.66_{0.03}$ | $0.39_{0.01}$ |
| | MCD | $0.75_{0.00}$ | $\underline{0.72_{0.00}}$ | $0.49_{0.00}$ | $0.63_{0.00}$ | $\underline{0.46_{0.00}}$ | $0.39_{0.00}$ |
| | ENS | $0.64_{0.02}$ | $\mathbf{0.71_{0.00}}$ | $0.45_{0.04}$ | $0.50_{0.03}$ | $0.57_{0.05}$ | $\underline{0.38_{0.02}}$ |
| | LAP | $2.75_{1.30}$ | $2.13_{1.00}$ | $1.08_{0.51}$ | $1.48_{0.70}$ | $1.62_{0.76}$ | $1.14_{0.54}$ |
| | BLoB | $\mathbf{0.61_{0.00}}$ | $0.73_{0.01}$ | $\underline{0.39_{0.00}}$ | $\underline{0.55_{0.00}}$ | $0.49_{0.00}$ | $0.39_{0.00}$ |
| | Ours | $0.64_{0.00}$ | $0.72_{0.00}$ | $\mathbf{0.35_{0.00}}$ | $\mathbf{0.46_{0.00}}$ | $\mathbf{0.44_{0.00}}$ | $\mathbf{0.37_{0.00}}$ |

the validation set into two parts with a 1:2 ratio for validation and testing. The checkpoint that performed best on the validation set was used for testing to obtain the final results.

We evaluated the effectiveness of Me-LoRA by measuring the accuracy (ACC) and negative log-likelihood (NLL). We expected calibration error (ECE) during the fine-tuning of LlaMA2-7B and LlaMA2-13B on commonsense reasoning tasks. We compared Me-LoRA with SOTA uncertainty estimation methods applied to the LoRA adapters of LLMs, including Maximum A Posteriori (MAP) with a weight decay rate of 1e2, Monte Carlo Dropout (MCD) with an ensemble size of 3 (using a dropout rate of 0.1 during fine-tuning) Gal & Ghahramani (2016), Deep Ensemble (ENS) Lakshminarayanan et al. (2017); Balabanov & Linander (2024); Wang et al. (2023) with three LoRA fine-tuned LLMs, Laplace-LoRA (LAP) Yang et al. (2023), and the latest BLoB Wang et al. (2024).

We re-implemented LAP and applied it to the MAP checkpoints. For BLoB, since no open-source code was available, we replicated the approach based on the description in the paper. To ensure a fair comparison, we made appropriate parameter adjustments. BLoB was only sampled once during each training, validation, and testing stage. The Flipout sampling technique and KL regularization from the original BLoB paper were not used in our replication, as they did not perform well. Instead, we applied the KL regularization method from Me-LoRA.

## 4.1 Performance on In-distribution Datasets

We evaluated with in-distribution fine-tuning Llama2-13B on six common sense reasoning tasks: Winogrande-small (WG-S) Sakaguchi et al. (2021), Winogrande-medium (WG-M) Sakaguchi et al. (2021), ARC-Challenge (ARC-C) Clark et al. (2018), ARC-Easy (ARC-E) Clark et al. (2018), Open-BookQA (OBQA) Mihaylov et al. (2018), and BoolQ Clark et al. (2019). We use the same pre-trained LLM backbone and datasets for all baseline methods, with additional validation sets used to select the final test checkpoint (Detailed settings can be found in the appendix B.2).

Table 2 shows the performance comparison of Me-LoRA and state-of-the-art models on the ACC, ECE, and NLL metrics on the test set with the pre-trained Llama2-13B model. MAP, MCD, ENS, and LAP exhibited specific overconfidence issues during the fine-tuning process. Compared to other models facing the challenge of uncertainty estimation during LLM fine-tuning, BLoB provides bet-

Table 3: Performance on in-distribution and out-of-distribution datasets. All the uncertainty estimation methods are applied to the LoRA adapter added upon the pre-trained Llama2-13B weights.

| Metric | Method | In-Dist. OBQA | Smaller Dist. Shift ARC-C | ARC-E | Larger Dist. Shift Chem. | Phy. | Math. |
|---|---|---|---|---|---|---|---|
| ACC ↑ | MAP | $83.80_{1.14}$ | $\mathbf{75.79_{0.84}}$ | $82.39_{0.50}$ | $38.33_{3.09}$ | $\mathbf{45.14_{2.60}}$ | $35.76_{0.49}$ |
| | MCD | $\underline{83.81_{0.31}}$ | $72.45_{0.05}$ | $81.02_{0.04}$ | $39.01_{0.25}$ | $\underline{37.49_{0.27}}$ | $35.97_{0.37}$ |
| | ENS | $83.33_{0.34}$ | $\underline{74.46_{0.90}}$ | $\mathbf{83.39_{0.49}}$ | $\underline{42.71_{3.07}}$ | $33.67_{0.47}$ | $35.07_{5.53}$ |
| | LAP | $82.56_{0.27}$ | $72.08_{0.08}$ | $82.13_{0.07}$ | $41.49_{0.69}$ | $32.24_{1.15}$ | $\mathbf{39.76_{0.22}}$ |
| | BLoB | $82.40_{0.13}$ | $72.27_{0.06}$ | $\underline{83.24_{0.14}}$ | $42.66_{0.10}$ | $37.14_{0.30}$ | $32.76_{0.41}$ |
| | Ours | $\mathbf{84.89_{0.08}}$ | $73.80_{0.13}$ | $82.58_{0.05}$ | $\mathbf{42.75_{0.08}}$ | $34.77_{0.42}$ | $33.65_{0.42}$ |
| ECE ↓ | MAP | $10.88_{0.78}$ | $15.11_{1.65}$ | $10.96_{0.43}$ | $30.00_{3.69}$ | $27.65_{5.10}$ | $30.79_{1.31}$ |
| | MCD | $\mathbf{3.82_{0.05}}$ | $\mathbf{5.56_{0.06}}$ | $\mathbf{1.90_{0.19}}$ | $20.09_{0.74}$ | $\mathbf{14.24_{0.37}}$ | $\underline{16.87_{0.88}}$ |
| | ENS | $10.56_{0.68}$ | $13.84_{0.48}$ | $8.21_{0.34}$ | $26.00_{3.86}$ | $29.76_{2.56}$ | $26.47_{4.85}$ |
| | LAP | $7.90_{0.13}$ | $10.39_{0.12}$ | $4.81_{0.16}$ | $20.80_{0.35}$ | $23.19_{1.48}$ | $16.95_{1.18}$ |
| | BLoB | $6.00_{0.14}$ | $9.73_{0.09}$ | $3.76_{0.19}$ | $\underline{17.76_{0.29}}$ | $18.67_{0.25}$ | $19.85_{0.89}$ |
| | Ours | $\underline{5.53_{0.27}}$ | $\underline{6.92_{0.09}}$ | $\underline{3.10_{0.09}}$ | $\mathbf{15.48_{0.56}}$ | $\underline{18.64_{0.70}}$ | $\mathbf{15.88_{0.45}}$ |
| NLL ↓ | MAP | $0.68_{0.05}$ | $0.90_{0.05}$ | $0.69_{0.04}$ | $1.90_{0.19}$ | $1.68_{0.12}$ | $1.91_{0.08}$ |
| | MCD | $\underline{0.46_{0.00}}$ | $\underline{0.70_{0.00}}$ | $\underline{0.49_{0.00}}$ | $\mathbf{1.26_{0.00}}$ | $\mathbf{1.36_{0.00}}$ | $\mathbf{1.48_{0.00}}$ |
| | ENS | $0.60_{0.02}$ | $0.84_{0.01}$ | $0.56_{0.01}$ | $1.59_{0.07}$ | $1.83_{0.08}$ | $1.72_{0.09}$ |
| | LAP | $1.66_{0.78}$ | $2.28_{1.07}$ | $1.51_{0.71}$ | $4.10_{1.92}$ | $4.61_{2.17}$ | $4.53_{2.14}$ |
| | BLoB | $0.49_{0.00}$ | $0.76_{0.01}$ | $\mathbf{0.47_{0.00}}$ | $\underline{1.31_{0.02}}$ | $1.52_{0.01}$ | $\underline{1.53_{0.01}}$ |
| | Ours | $\mathbf{0.44_{0.00}}$ | $\mathbf{0.68_{0.00}}$ | $\underline{0.49_{0.00}}$ | $\mathbf{1.26_{0.00}}$ | $\underline{1.41_{0.00}}$ | $\mathbf{1.48_{0.00}}$ |

ter uncertainty estimation performance and mitigates overconfidence. However, with an improved ability to quantify uncertainty, the model parameters also increase significantly compared to LoRA.

Me-LoRA achieves better or comparable performance across all datasets. With a single sampling, Me-LoRA provides superior uncertainty estimation performance and significantly mitigates overconfidence while maintaining comparable or better ACC, NLL, and ECE. Surprisingly, using only the same number of parameters as LoRA, Me-LoRA achieves significantly higher ACC than other baselines on most datasets (ARC-C, ARC-E, WG-M, OBQA) without a substantial loss in uncertainty estimation quality. This observation confirms that jointly learning the mean and covariance of low-rank matrices during fine-tuning can mutually improve their quality.

In addition, we also conducted tests on Llama2-7B, with detailed results in the appendix B.3. Our method maintains similar or even better performance than the state-of-the-art models on different datasets, demonstrating its generalization.

## 4.2 PERFORMANCE ON OUT-OF-DISTRIBUTION DATASETS

We fine-tune models on the OBQA dataset Mihaylov et al. (2018), which consists of multiple-choice elementary-level science questions, to assess the generalization ability of various methods under distributional shifts. We categorize shifts between datasets into two types: smaller and larger. The ARC dataset Clark et al. (2018), also composed of multiple-choice science questions, represents a smaller distributional shift. In contrast, the college-level chemistry, physics, and mathematics subsets from MMLU Hendrycks et al. (2020a;b) serve as examples of larger distributional shifts.

Table 3 highlights Me-LoRA's comparable out-of-distribution (OOD) generalization ability compared to other methods on datasets with varying distributions. Me-LoRA exhibits comparable ACC on out-of-distribution datasets, especially achieving the highest accuracy on the chemistry subset. By incorporating uncertainty through sampling, Me-LoRA enhances the generalization capability. Regarding uncertainty estimation, Me-LoRA demonstrates either the best or second-best performance under smaller and larger distribution shifts.

## 5 RELATED WORKS

The rapid development of LLMs has intensified research on efficient fine-tuning for specific tasks. Traditional full-parameter fine-tuning is computationally expensive and presents challenges related to storage and deployment. Consequently, PEFT methods have been investigated to minimize computational and storage resource requirements while preserving model performance. LoRA (Hu et al., 2021) is a prominent PEFT method that injects trainable low-rank matrices at each layer of Transformer architecture to approximate weight changes, effectively reducing the number of trainable parameters and enhancing training efficiency without increasing inference latency. Additionally, several other PEFT methods have been developed, including adapter-based fine-tuning (Rücklé et al., 2020; Pfeiffer et al., 2020), prompt-based fine-tuning (Lester et al., 2021; Vu et al., 2021; Asai et al., 2022), and partial fine-tuning (Ansell et al., 2021; Zaken et al., 2021). These approaches aim to minimize parameter count while maintaining or improving model performance.

Overconfidence in LLM predictions can lead to risks, highlighting the importance of uncertainty estimation for enhancing model reliability and trustworthiness. However, existing LLMs often exhibit overconfidence after fine-tuning, making it challenging to estimate predictive uncertainty accurately. To address this issue, various uncertainty estimation techniques have been introduced in LLMs, including Monte Carlo dropout (Gal & Ghahramani, 2016), deep ensemble (Lakshminarayanan et al., 2017; Wang et al., 2023; Zhai et al., 2023), and Laplace approximation(Antorán et al., 2022; Yang et al., 2023). These methods enhance the model's uncertainty estimation capabilities by incorporating randomness during training or establishing probability distributions over model parameters. A notable approach is BLoB (Wang et al., 2024), which integrates Bayesian inference with LoRA to capture model uncertainty through the distribution of low-rank adaptation parameters. In contrast to the post-hoc method Laplace-LoRA (Yang et al., 2023), BLoB updates both the means and covariances of LLM parameters during fine-tuning, facilitating more accurate uncertainty estimation.

Table 4: Performance of ablation methods on Llama2-13B pre-trained weights. The evaluation is done across six common-sense reasoning tasks with a shared hyper-parameter setting after 5,000 training steps. w/o noise: do not add noise $\varepsilon$ to the prior variance $\sigma_p$

| Metric | Method | WG-S | ARC-C | ARC-E | WG-M | OBQA | BoolQ |
|---|---|---|---|---|---|---|---|
| ACC ↑ | w/o noise | $\textbf{69.83}_{\textbf{0.08}}$ | $70.89_{0.05}$ | $87.01_{0.05}$ | $74.08_{0.15}$ | $\textbf{83.83}_{\textbf{0.11}}$ | $\textbf{82.66}_{\textbf{0.19}}$ |
|  | Ours | $68.67_{0.07}$ | $\textbf{71.59}_{\textbf{0.11}}$ | $\textbf{87.11}_{\textbf{0.06}}$ | $\textbf{75.66}_{\textbf{0.09}}$ | $82.06_{0.27}$ | $82.55_{0.03}$ |
| ECE ↓ | w/o noise | $\textbf{8.41}_{\textbf{0.09}}$ | $10.72_{0.07}$ | $\textbf{3.94}_{\textbf{0.06}}$ | $\textbf{1.72}_{\textbf{0.09}}$ | $5.02_{0.40}$ | $\textbf{1.63}_{\textbf{0.47}}$ |
|  | Ours | $9.25_{0.06}$ | $\textbf{5.06}_{\textbf{0.08}}$ | $4.29_{0.06}$ | $4.23_{0.18}$ | $\textbf{2.78}_{\textbf{0.23}}$ | $2.28_{0.10}$ |
| NLL ↓ | w/o noise | $\textbf{0.62}_{\textbf{0.00}}$ | $0.80_{0.00}$ | $\textbf{0.41}_{\textbf{0.00}}$ | $0.52_{0.00}$ | $0.48_{0.01}$ | $\textbf{0.39}_{\textbf{0.00}}$ |
|  | Ours | $0.64_{0.06}$ | $\textbf{0.71}_{\textbf{0.08}}$ | $0.42_{0.00}$ | $\textbf{0.51}_{\textbf{0.00}}$ | $\textbf{0.47}_{\textbf{0.00}}$ | $0.40_{0.00}$ |

Table 5: Performance of ablation methods on in-distribution and out-of-distribution datasets.

| Metric | Method | In-Dist. OBQA | Smaller Dist. Shift | | Larger Dist. Shift | | |
|---|---|---|---|---|---|---|---|
|  |  |  | ARC-C | ARC-E | Chem. | Phy. | Math. |
| ACC ↑ | w/o noise | $\textbf{83.17}_{\textbf{0.22}}$ | $\textbf{72.87}_{\textbf{0.15}}$ | $\textbf{83.57}_{\textbf{0.15}}$ | $40.94_{0.20}$ | $32.92_{0.61}$ | $37.13_{1.29}$ |
|  | Ours | $82.35_{0.13}$ | $71.45_{0.16}$ | $82.95_{0.06}$ | $\textbf{41.59}_{\textbf{0.62}}$ | $\textbf{33.45}_{\textbf{0.51}}$ | $\textbf{39.30}_{\textbf{0.47}}$ |
| ECE ↓ | w/o noise | $\textbf{5.21}_{\textbf{0.21}}$ | $\textbf{8.24}_{\textbf{0.20}}$ | $\textbf{2.98}_{\textbf{0.11}}$ | $17.34_{0.18}$ | $20.70_{1.07}$ | $15.29_{0.94}$ |
|  | Ours | $6.17_{0.29}$ | $9.37_{0.13}$ | $3.78_{0.22}$ | $\textbf{15.78}_{\textbf{0.89}}$ | $\textbf{18.25}_{\textbf{0.65}}$ | $\textbf{14.40}_{\textbf{0.57}}$ |
| NLL ↓ | w/o noise | $\textbf{0.48}_{\textbf{0.00}}$ | $\textbf{0.74}_{\textbf{0.00}}$ | $\textbf{0.46}_{\textbf{0.00}}$ | $1.32_{0.01}$ | $1.49_{0.02}$ | $1.51_{0.01}$ |
|  | Ours | $0.49_{0.01}$ | $0.75_{0.00}$ | $0.48_{0.00}$ | $\textbf{1.29}_{\textbf{0.01}}$ | $\textbf{1.43}_{\textbf{0.01}}$ | $\textbf{1.46}_{\textbf{0.01}}$ |

## 6 ABLATION STUDY

In this section, we conduct an ablation study to examine the impact of introducing noise $\varepsilon$ into the prior variance. All subsequent experiments focus on both in-distribution and out-of-distribution

datasets using the Llama2-13B model. We maintain the same hyperparameters as in previous experiments, the only modification being whether noise is added to the prior variance.

Table 4 and Table 5 show that the method without added noise performs better than ours on WG-S, OBQA, and BoolQ. However, our approach significantly outperforms the noise-free method on out-of-distribution tasks, particularly in college mathematics, physics, and chemistry. This problem indicates that adding noise helps mitigate overfitting and enhances generalization capabilities. The ablation study further demonstrates that while both methods maintain similar generalization performance, the noise-augmented method consistently achieves higher accuracy, underscoring the benefits of incorporating noise.

## 7 CONCLUSION

We propose Me-LoRA, a novel parameter-efficient Bayesian fine-tuning method for LLMs in this work. Our method shows that a full-weight variational distribution can be efficiently optimized by a low-dimensional square matrix incorporating a variance estimate positioned between the two low-rank matrices in LoRA. In our experiments, we observed enhanced generalization and uncertainty estimation performance compared with several baseline methods. Our approach highlights that jointly learning the mean and covariance of variational distributions with a small number of parameters during fine-tuning can improve each other, greatly enhancing the efficiency of Bayesian methods.

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

# A   VARIATIONAL DISTRIBUTION OF THE FULL-WEIGHT MATRIX

With the pre-trained weight matrix $\boldsymbol{W}_0 \in \mathbb{R}^{m \times n}$, the low-rank weight update matrix $\boldsymbol{A} \in \mathbb{R}^{r \times n}$ and $\boldsymbol{B} \in \mathbb{R}^{m \times r}$, we suppose that the variational distribution of the other low-rank update matrix $\boldsymbol{C} \in \mathbb{R}^{r \times r}$ is Gaussian with mean $\boldsymbol{M}$ and standard deviation $\boldsymbol{\Omega}$, denoted as $q(\boldsymbol{C}) \sim \mathcal{N}(\boldsymbol{M}, \boldsymbol{\Omega})$, where $\boldsymbol{M} \in \mathbb{R}^{r \times r}$ and $\boldsymbol{\Omega} \in \mathbb{R}^{r \times r}$. We then have $q(\boldsymbol{Q}) \sim \mathcal{N}(\boldsymbol{MA}, \boldsymbol{\Omega}|\boldsymbol{A}|)$, where $\boldsymbol{Q} = \boldsymbol{CA}$. The equivalent variational distribution defined on the full weight matrix $\boldsymbol{W} = \boldsymbol{W}_0 + \boldsymbol{BCA}$ as fallow,

$$q(\boldsymbol{W}) = \mathcal{N}(\boldsymbol{W}; \boldsymbol{\mu}_q, \boldsymbol{\Sigma}_q), \tag{9}$$

where $\boldsymbol{\mu}_q = \text{vec}(\boldsymbol{W}_0 + \boldsymbol{BMA})$ and $\boldsymbol{\Sigma}_q = [\boldsymbol{I} \otimes \boldsymbol{B}] \cdot [\text{diag}(\text{vec}(\boldsymbol{\Omega}|\boldsymbol{A}|)^2)] \cdot [\boldsymbol{I} \otimes \boldsymbol{B}^\top]$

We begin by calculating the mean value of $q(\boldsymbol{W})$,

$$\begin{aligned} \mu_q &= \text{vec}(\mathbb{E}[\boldsymbol{W}_0 + \boldsymbol{BCA}]) \\ &= \text{vec}(\boldsymbol{W}_0 + \boldsymbol{B}\mathbb{E}[\boldsymbol{C}]\boldsymbol{A}) \\ &= \text{vec}(\boldsymbol{W}_0 + \boldsymbol{BMA}). \end{aligned} \tag{10}$$

We then calculate the covariance matrix $\boldsymbol{\Sigma}_q$ as

$$\begin{aligned} \boldsymbol{\Sigma}_q &= \mathbb{E}\left[ (\text{vec}(\boldsymbol{W}) - \mathbb{E}[\text{vec}(\boldsymbol{W})]) \cdot (\text{vec}(\boldsymbol{W}) - \mathbb{E}[\text{vec}(\boldsymbol{W})])^\top \right] \\ &= \mathbb{E}\left[ \text{vec}(\boldsymbol{B}(\boldsymbol{C} - \boldsymbol{M})\boldsymbol{A}) \cdot \text{vec}(\boldsymbol{B}(\boldsymbol{C} - \boldsymbol{M})\boldsymbol{A})^\top \right] \\ &= \mathbb{E}\left[ [\boldsymbol{I} \otimes \boldsymbol{B}] \cdot \text{vec}(\boldsymbol{CA} - \boldsymbol{MA}) \cdot \text{vec}(\boldsymbol{CA} - \boldsymbol{MA})^\top \cdot [\boldsymbol{I} \otimes \boldsymbol{B}]^\top \right] \\ &= [\boldsymbol{I} \otimes \boldsymbol{B}]\mathbb{E}\left[ \text{vec}(\boldsymbol{CA} - \boldsymbol{MA}) \cdot \text{vec}(\boldsymbol{CA} - \boldsymbol{MA})^\top \right] [\boldsymbol{I} \otimes \boldsymbol{B}]^\top \\ &= [\boldsymbol{I} \otimes \boldsymbol{B}] \cdot [\text{diag}(\text{vec}(\boldsymbol{\Omega}|\boldsymbol{A}|)^2)] \cdot [\boldsymbol{I} \otimes \boldsymbol{B}]^\top \end{aligned} \tag{11}$$

Table 6: Comparison of running time and memory cost of BLoB finetuning for Llama2-7B and Llama2-13B. The evaluation is based on fine-tuning for 5,000 steps.

| Model | Metric | Method | Datasets | | | | | |
|-------|--------|--------|------|------|-------|-------|------|-------|
| | | | WG-S | WG-M | ARC-C | ARC-E | OBQA | BoolQ |
| Llama2-7B | Time (Seconds) ↓ | LoRA | 1731 | 2824 | 2590 | 1718 | 2151 | 11070 |
| | | BLoB | 2475 | 4395 | 3601 | 2479 | 3095 | 15267 |
| | | Me-LoRA | 2699 | 4214 | 3972 | 2697 | 3297 | 16840 |
| | Memory (GB) ↓ | LoRA | 4.80 | 8.16 | 8.38 | 4.93 | 6.73 | 7.30 |
| | | BLoB | 4.82 | 8.17 | 8.40 | 4.94 | 6.75 | 7.27 |
| | | Me-LoRA | 4.80 | 8.19 | 8.39 | 4.94 | 6.74 | 7.28 |
| Llama2-13B | Time (Seconds) ↓ | LoRA | 2230 | 3626 | 3312 | 2231 | 2667 | 14038 |
| | | BLoB | 3401 | 5396 | 4908 | 3423 | 4044 | 15244 |
| | | Me-LoRA | 3256 | 5470 | 4314 | 3258 | 4098 | 21255 |
| | Memory (GB) ↓ | LoRA | 8.80 | 13.99 | 14.28 | 8.94 | 11.78 | 12.66 |
| | | BLoB | 8.86 | 14.04 | 14.37 | 9.01 | 11.84 | 12.71 |
| | | Me-LoRA | 8.82 | 13.96 | 14.29 | 8.96 | 11.80 | 12.67 |

# B   SUPPLEMENTARY EXPERIMENTAL RESULTS

This section presents supplementary experimental results that were excluded from the main text due to space constraints. In Appendix B.1, we first report the memory and training time requirements of Me-LoRA. Appendix B.2 provides a detailed analysis of the memory and training time requirements. Finally, in Appendix B.3, we subsequently perform an ablation study focused on the noise component in the prior of Me-LoRA.

## B.1 MEMORY AND TRAINING TIME REQUIREMENTS

By introducing an additional standard deviation matrix $\Omega$, which has the same dimensions as the LoRA $A$ matrix, the number of trainable parameters in BLoB increases by approximately 50% compared to LoRA. In contrast, the number of trainable parameters in Me-LoRA only increases by less than 0.2%. However, the computation of the KL divergence, along with the inclusion of the additional standard deviation matrix in the likelihood loss, results in extra time required for both forward and backward passes. We conducted experiments using an NVIDIA RTX 3090 GPU to train the Llama2-7B model, and an NVIDIA RTX A40 GPU to train the Llama2-13B model, to evaluate the differences in GPU memory consumption and training time across BLoB, Me-LoRA, and standard LoRA fine-tuning. The results are shown in Table 6.

Using LoRA as a baseline, on Llama2-7B, BLoB increased memory consumption by approximately 0.12% and training time by about 40%, while Me-LoRA increased memory consumption by approximately 0.09% and training time by about 50%. For Llama2-13B, BLoB increased memory usage by approximately 0.54% and training time by around 30%, whereas Me-LoRA increased memory consumption by about 0.07% and training time by around 48%. Although our method outperforms BLoB in terms of memory efficiency, it lags behind in terms of training time. We hypothesize that this is due to the time complexity of the matrix multiplication between $A$ and $C$, which takes longer than the element-wise addition of matrix $A$ and its variance in BLoB.

## B.2 HYPERPARAMETERS

Table 7: Hyperparameters of LoRA and Me-LoRA-Specific Hyperparameters.

| Hyperparameter | Llama2-7B      Llama2-13B |
|---|---|
| Optimizer | AdamW |
| LR Scheduler | Linear |
| Warmup Ratio | 0.02 |
| Learning Rate | $1 \times 10^{-4}$ |
| Dropout Probability | 0.1 |
| Batch Size | 4 |
| Max Seq. Len. | 300 |
| LoRA $\alpha$ | 16 |
| LoRA $r$ | 8 |
| Optimizer of KL | SGD |
| LR of KL | $1 \times 10^{-4}$ |
| $\sigma_p$ | 0.2 |

## B.3 PERFORMANCE ON LLAMA2-7B

We also conducted tests on Llama2-7B model. The results on in-distribution and out-of-distribution datasets are shown in Tables 8 and 9, respectively. we obtained comparable results on the in-distribution datasets, even some of which are the best or the second best. However, we did not achieve satisfactory results on the out-of-distribution datasets, and slight overfitting was observed. We hypothesize that this may be due to limitations in the model's ability to generalize beyond the training data. Specifically, the current normalization and regularization techniques may not be sufficiently robust to handle the variations present in unseen data, suggesting that improvements in these areas could enhance the model's performance on out-of-distribution tasks. Future work will focus on refining these mechanisms to mitigate overfitting and improve generalization.

Table 8: Performance of different methods applied to LoRA on Llama2-7B pre-trained weights. The evaluation is done across six common-sense reasoning tasks with a shared hyper-parameter setting after 5,000 training steps.

| Metric | Method | WG-S | ARC-C | ARC-E | WG-M | OBQA | BoolQ |
|---|---|---|---|---|---|---|---|
| ACC ↑ | MAP | $67.18_{0.88}$ | $66.72_{1.01}$ | $84.65_{0.32}$ | $73.72_{0.40}$ | $81.73_{0.18}$ | $80.63_{0.28}$ |
| | MCD | $65.89_{0.03}$ | $64.31_{0.06}$ | $84.81_{0.04}$ | $73.58_{0.03}$ | $79.82_{0.09}$ | $81.77_{0.11}$ |
| | ENS | $66.06_{0.09}$ | $63.08_{0.92}$ | $84.68_{0.41}$ | $72.90_{0.49}$ | $80.93_{0.52}$ | $80.31_{1.06}$ |
| | LAP | $64.61_{0.25}$ | $65.26_{0.06}$ | $84.00_{0.09}$ | $71.55_{0.08}$ | $80.52_{0.08}$ | $80.51_{0.03}$ |
| | BLoB | $66.71_{0.17}$ | $65.06_{0.06}$ | $84.84_{0.12}$ | $72.36_{0.29}$ | $76.94_{0.24}$ | $77.34_{0.12}$ |
| | Ours | $66.15_{0.04}$ | $65.62_{0.05}$ | $84.93_{0.03}$ | $72.10_{0.01}$ | $81.78_{0.03}$ | $81.61_{0.04}$ |
| ECE ↓ | MAP | $29.39_{1.28}$ | $22.40_{2.33}$ | $10.77_{0.51}$ | $17.62_{2.09}$ | $7.92_{2.09}$ | $6.38_{1.54}$ |
| | MCD | $17.97_{0.03}$ | $6.45_{0.09}$ | $9.81_{0.04}$ | $8.70_{0.04}$ | $11.89_{0.21}$ | $2.34_{0.11}$ |
| | ENS | $14.20_{5.24}$ | $6.12_{3.12}$ | $8.95_{0.65}$ | $9.60_{2.05}$ | $4.48_{1.10}$ | $5.92_{1.04}$ |
| | LAP | $5.25_{0.09}$ | $15.83_{0.08}$ | $11.24_{0.08}$ | $9.03_{0.17}$ | $5.40_{0.70}$ | $1.54_{0.11}$ |
| | BLoB | $7.64_{0.15}$ | $8.09_{0.09}$ | $6.44_{0.15}$ | $11.04_{0.24}$ | $6.24_{0.15}$ | $2.35_{0.15}$ |
| | Ours | $9.13_{0.09}$ | $8.69_{0.16}$ | $7.08_{0.14}$ | $6.45_{0.07}$ | $4.87_{0.06}$ | $5.05_{0.06}$ |
| NLL ↓ | MAP | $1.99_{0.36}$ | $1.36_{0.18}$ | $0.67_{0.04}$ | $0.90_{0.15}$ | $0.55_{0.04}$ | $0.45_{0.01}$ |
| | MCD | $0.81_{0.00}$ | $0.89_{0.00}$ | $0.62_{0.00}$ | $0.59_{0.00}$ | $0.69_{0.01}$ | $0.41_{0.00}$ |
| | ENS | $0.75_{0.09}$ | $0.95_{0.04}$ | $0.58_{0.03}$ | $0.62_{0.05}$ | $0.53_{0.01}$ | $0.46_{0.02}$ |
| | LAP | $1.90_{0.90}$ | $3.04_{1.44}$ | $2.21_{1.05}$ | $1.81_{0.85}$ | $1.77_{0.83}$ | $1.29_{0.61}$ |
| | BLoB | $0.64_{0.00}$ | $0.91_{0.00}$ | $0.52_{0.01}$ | $0.65_{0.01}$ | $0.64_{0.01}$ | $0.48_{0.00}$ |
| | Ours | $0.66_{0.07}$ | $0.89_{0.00}$ | $0.55_{0.00}$ | $0.59_{0.00}$ | $0.51_{0.00}$ | $0.42_{0.00}$ |

Table 9: Performance on in-distribution and out-of-distribution datasets. All the uncertainty estimation methods are applied to the LoRA adapter added upon the pre-trained Llama2-7B weights.

| Metric | Method | In-Dist. OBQA | Smaller Dist. Shift ARC-C | ARC-E | Larger Dist. Shift Chem. | Phy. | Math. |
|---|---|---|---|---|---|---|---|
| ACC ↑ | MAP | $81.73_{0.19}$ | $66.52_{0.56}$ | $76.96_{0.04}$ | $39.58_{2.55}$ | $26.00_{1.63}$ | $36.46_{0.85}$ |
| | MCD | $78.75_{0.14}$ | $65.21_{0.15}$ | $72.52_{0.03}$ | $37.86_{0.71}$ | $28.77_{0.29}$ | $36.71_{0.33}$ |
| | ENS | $79.53_{1.70}$ | $65.53_{1.09}$ | $75.42_{0.69}$ | $35.76_{2.73}$ | $25.00_{1.41}$ | $36.46_{0.85}$ |
| | LAP | $80.64_{0.22}$ | $65.61_{0.10}$ | $76.87_{0.11}$ | $38.03_{0.52}$ | $24.60_{0.33}$ | $39.57_{0.19}$ |
| | BLoB | $77.05_{0.25}$ | $60.81_{0.11}$ | $74.31_{0.10}$ | $30.84_{0.35}$ | $23.62_{0.37}$ | $38.01_{0.54}$ |
| | Ours | $81.80_{0.05}$ | $65.57_{0.03}$ | $76.59_{0.04}$ | $33.40_{0.14}$ | $24.74_{0.33}$ | $34.50_{0.15}$ |
| ECE ↓ | MAP | $7.92_{2.09}$ | $14.97_{2.59}$ | $9.36_{2.38}$ | $21.42_{1.60}$ | $31.82_{4.20}$ | $25.69_{5.11}$ |
| | MCD | $3.83_{0.33}$ | $7.36_{0.28}$ | $3.96_{0.03}$ | $10.87_{0.81}$ | $19.97_{0.19}$ | $16.29_{0.64}$ |
| | ENS | $7.69_{3.78}$ | $9.56_{1.44}$ | $5.53_{1.22}$ | $16.73_{1.26}$ | $27.35_{2.15}$ | $14.49_{2.24}$ |
| | LAP | $5.33_{0.60}$ | $14.51_{0.09}$ | $9.10_{0.07}$ | $17.57_{0.48}$ | $32.56_{0.28}$ | $15.52_{0.90}$ |
| | BLoB | $7.11_{0.26}$ | $17.08_{0.08}$ | $9.29_{0.09}$ | $29.73_{0.42}$ | $32.60_{0.39}$ | $18.36_{0.21}$ |
| | Ours | $4.82_{0.07}$ | $11.71_{0.02}$ | $6.78_{0.04}$ | $20.56_{0.44}$ | $29.73_{0.37}$ | $18.58_{0.44}$ |
| NLL ↓ | MAP | $0.55_{0.04}$ | $0.96_{0.06}$ | $0.68_{0.04}$ | $1.40_{0.02}$ | $1.70_{0.12}$ | $1.56_{0.11}$ |
| | MCD | $0.58_{0.00}$ | $0.88_{0.00}$ | $0.63_{0.00}$ | $1.30_{0.01}$ | $1.48_{0.01}$ | $1.39_{0.00}$ |
| | ENS | $0.59_{0.07}$ | $0.89_{0.03}$ | $0.67_{0.03}$ | $1.34_{0.03}$ | $1.54_{0.06}$ | $1.45_{0.04}$ |
| | LAP | $1.74_{0.83}$ | $2.97_{1.40}$ | $2.06_{0.97}$ | $4.17_{1.96}$ | $4.81_{2.26}$ | $4.38_{2.07}$ |
| | BLoB | $0.63_{0.01}$ | $1.09_{0.00}$ | $0.72_{0.00}$ | $1.64_{0.02}$ | $1.77_{0.01}$ | $1.54_{0.03}$ |
| | Ours | $0.51_{0.00}$ | $0.91_{0.00}$ | $0.64_{0.00}$ | $1.37_{0.00}$ | $1.68_{0.00}$ | $1.45_{0.00}$ |

