# OpenReview forum: "ME-LORA: MEMORY-EFFICIENT BAYESIAN LOW- RANK ADAPTATION FOR LARGE LANGUAGE MODELS"
_ICLR.cc/2025/Conference — ICLR 2025 Conference Withdrawn Submission_

### Official Review · Reviewer_XLS9 · 2024-10-24

**Soundness:** 3
**Presentation:** 2
**Contribution:** 2
**Rating:** 3
**Confidence:** 4

**Summary:**

The paper is built upon a recent work BLoB, which applies black box variational inference on LoRA during fine-tuning. This paper suggests a new method: Me-LoRA, which improves upon BLOB in terms of parameter counting, in particular, instead of having a variational distribution directly over LoRA's A and B components, which has a total number of parameters as 2 * (r * n + r* m), it introduces a new components C of shape r by r and perform VI on C instead of directly on A and B, as such the total number of parameters is reduced to r * n + r* m + 2 * r * r, and helps reduce the number of parameters significantly when r << n, which is the case for most LLM. The model then shows that the proposed approach shows comparable / slightly better than the past approach BLOB, but smaller parameter overhead.

Overall, I find the method technically sound and the experiments fairly convincing, the proposed modification to the existing approach is well-motivated and can have some usefulness. However, the writing needs some significant improvement (see weakness).

**Strengths:**

- The proposed method is easy and technically sound.

- The closed-form computation of KL divergence is carefully derived.

- The experiments are conducted on a wide range of tasks, although mostly multiple choice QA problems, to demonstrate the effectiveness of the problem.

- A reasonable amount of experiment details are provided for reproducibility (though code not provided along with the submission).

**Weaknesses:**

- It would be nice if the references can be colored;

- End of line 111 is missing parathesis?

- The description of Eq.3 is incorrect, it is the KL divergence between the variational posterior and the prior not the posterior. Also it's more common to call it ELBO rather than free energy in Bayesian deep learning literature

- What is \theta in Eq. (3)?

- The definition of q(C) at line 178 is confusing: If M is a matrix, then q(C) should be a matrix normal, how could it have a r by r matrix as the covariance?

- The math language is also not consistent: Eq (4) has W; in \mathcal{N}, but q(C) does not.

- Having a randomized prior is extremely weird and non-standard (Sec 3.3 , Eq. 8), it's also not stated what is U(0, 1), which I guess is a uniform distribution.

- It would be nice if the parameter count section can be summarized into a table for easier comparison.

- Line 267, ' ..saved model checkpoints every 100 steps,...' it's nice to have experiment details presented but I don't think this piece of information is necessary.

- Line 398, the authors mentioned Flipout, but did not provide reference nor any explanation.

- Table 4 and 5 being put at the bottom of related work section is weird.

**Questions:**

- It should be clarified that A, and B are not variational parameters, they do not fit into the ELBO defined in Eq. (3). They are often referred to as "model hyper-parameter".

- Section 3.1 demonstrates the induced covariance matrix over the full-weight matrices, but does not go deeper into:
1. Is that covariance matrix diagonal, low-rank, or of any structure;
2. Why shall we care about this quantity? does the covariance matrix help us, e.g. better understand the landscape of LoRA fine-tuning, etc?

- Has the author considered using Monte Carlo estimation to estimate the KL divergence rather than using closed form solution?

- When performing VI, do the authors additionally utilize regularization such as weight decay or L2 regularization?

- Why the prior is set on the full model weights rather than just on the low-rank components (Eq. 5)?

- Why is ensemble worse than MAP in terms of accuracy? Does it mean ensembling could harm the performance potentially?

- Isn't a weight decay of 1e2 too large?

- Why is the proposed method suboptimal in ECE?

- How does the proposed method compare with Rank-1 BNN [1]?

- How many Monte Carlo samples are used for estimating the Bayesian model averaging?

- Why the numbers in the first column of Table 3 different from the numbers in Table 2.?

- Why do we need the random noise epsilon on the prior? The results in Table. 4 seem to be mixed.

- What benefits can we get from this approach, if we are in an open-ended generation setting? A huge body of LLM's applications are open-ended generation tasks such as translation, summarization, etc.

[1] Efficient and Scalable Bayesian Neural Nets with Rank-1 Factors

---

### Official Review · Reviewer_oTvV · 2024-10-29

**Soundness:** 3
**Presentation:** 3
**Contribution:** 3
**Rating:** 6
**Confidence:** 3

**Summary:**

The paper proposes a somewhat more efficient approach for Bayesian LoRA in LLMs.

**Strengths:**

The paper is well written, and has extensive experiments demonstrating that their method performs reasonably well.

**Weaknesses:**

One weakness is in the odd presentation of Bayesian LoRA in LLMs.  It would be far more preferable to present a historical overview, saying something like

>Yang et al. 2023 [or other earlier work] introduced the notion of doing Bayesian inference over the low-rank adapters for fine-tuning LLMs.  This had numerous advantages ... . However, Laplace inference, as used there had disadvantages ... . These disadvantages motivated the introduction of BLoB, which uses VI.  We build on BLoB ...

Me-LoRA only does Bayesian inference over C, and does MAP over A and B, which will likely reduce the benefits you might see from a fully Bayesian approach, and make it resemble more closely a non-Bayesian approach.

**Questions:**

How much do we care about the relatively modest reductions in memory usage in Table 1, as compared to the very large memory cost of the model itself?

---

### Official Review · Reviewer_mKt2 · 2024-10-31

**Soundness:** 2
**Presentation:** 2
**Contribution:** 2
**Rating:** 3
**Confidence:** 3

**Summary:**

This paper explores Bayesian Low-Rank Adaptation (LoRA), a method known to reduce overconfidence in inference when data is limited. The authors introduce a memory-efficient variant, Me-LoRA, by performing sampling on a small-scale intermediate matrix rather than the low-rank matrix directly. Experimental results with LLaMA2 models demonstrate that Me-LoRA maintains both effectiveness and efficiency compared to the original Bayesian LoRA framework.

**Strengths:**

1. The core idea of this paper is well-presented with a clear comparison against the original Bayesian LoRA framework.

2. Comprehensive experiments demonstrate that Me-LoRA achieves a balance between effectiveness and efficiency when compared to state-of-the-art methods.

**Weaknesses:**

1. The motivation behind the research problem is not clearly presented. The submission lacks an explanation of when overconfidence occurs in LLM inference, why this issue is critical, and how such overconfidence impacts the model's responses? These questions should be properly addressed.

2. Essentially, Me-LoRA is an efficient variant of BLoB and is supposed to replicate BLoB's performance with reduced resource demands. However, it appears to fall short in terms of ECE, a key metric for assessing overconfidence, against BLoB.

3. The computational cost comparison in Table 6 is confusing. The backbone model requires at least 13GB (LLaMA2-7B) or 21GB (LLaMA2-13B) GPU memory, yet the memory usage reported in Table 6 is significantly lower than that of the backbone model. Additionally, the rank used in the efficiency comparison is missing.

4. This submission seems to be incomplete considering the presented contents and presentation itself.  Further revisions are recommended to enhance its clarity and comprehensiveness.

**Questions:**

See Weaknesses.

---

### Official Review · Reviewer_irBx · 2024-11-01

**Soundness:** 2
**Presentation:** 3
**Contribution:** 2
**Rating:** 3
**Confidence:** 4

**Summary:**

The authors propose a small variant to the recent LoRA variant Bayesian Low-Rank Adaptation by Backpropagation (BLoB).  BLoB adapts a variational Bayesian setting wherein the LoRA parameters of the A matrix are parameterized by Gaussian priors.  Subsequently, BLoB makes the evaluation of the variational objective (i.e., the likelihood regularized by a KL-divergence term) efficient in practice by deriving the KL-divergence under assumed Gaussian priors, as well as incorporating flipout into LoRA for efficient sampling.

The introduced method, called ME-LoRA, near-directly adapts the BLoB framework.  The main technical difference is the use of a full matrix C of rank r (the lower dimension), which acts as an intermediate matrix which is multiplied between the LoRA B and A matrices, i.e., W = W_0 + BCA.  While BLoB includes one Gaussian per each value of the A matrix (leading to two learnable matrices of size r x n representing the Gaussians means and variances), ME-LoRA instead utilizes two learnable matrices of size r x r.  The authors attempt to reproduce both the BLoB framework and experimental set up from the BLoB, with ME-LoRA performing favorably on accuracy and negative likelihood tasks.

**Strengths:**

The proposed method is straight forward and the computational savings, compared to BLoB, are immediately obvious.  It is also commendable that the authors have undertaken the task of reproducing the methods and experiments from the original BLoB paper.

**Weaknesses:**

The proposed changes to BLoB are small contributions, which limit the potential impact of the work.  There are also several important concerns, in particular:
- In BLOB, for A \in R ^ {r \cross n}, each element of A is assumed to be an independent Gaussian, which is why the joint density is a product of the Gaussians.  However, in your setup (line 176):
> C ∈ Rr×r is Gaussian with mean M and standard deviation Ω, denoted as q(C) ∼ N (M, Ω),

which would mean the distribution on line 177, i.e., q(Q) \sim N(MA, \Sigma | A|), is incorrect.  For Q=CA, it should be
q(Q) \sim N(MA, A^T \Sigma A).  Why is there this discrepancy, and what does this mean for the results?
- Most importantly, there are concerns regarding the degree to which the authors' were able to faithfully reproduce both BLoB and the experiments of the BLoB paper.  Firstly, the results in Table 2 are significantly different than the BLoB paper (in fact, BLoB no longer state of the art on the majority of tasks).  Secondly, key ingredients of the BLoB paper did not work under reimplementation, as noted on lines 305-301:
> We re-implemented LAP and applied it to the MAP checkpoints. For BLoB, since no open-source
code was available, we replicated the approach based on the description in the paper. To ensure a
fair comparison, we made appropriate parameter adjustments. BLoB was only sampled once during
each training, validation, and testing stage. The Flipout sampling technique and KL regularization
from the original BLoB paper were not used in our replication, as they did not perform well. Instead,
we applied the KL regularization method from Me-LoRA.

As previously noted, it is commendable that the authors sought to reimplement the results from the BLoB paper, although the correctness of the reimplementation is a major concern.  With the release of the BLoB code, I would hope the authors could better reproduce the experimental set up from that paper and better incorporate their method into the official BloB source (with Flipout and KL regularization).
Official BLoB source (I understand this was just uploaded recently, I hope this aids the authors in their future efforts):
https://github.com/Wang-ML-Lab/bayesian-peft

**Questions:**

Please see above for major concerns.  Some minor comments and questions:

- In Section 3, it would be better to recap BLOB's LoRA framework, then discuss the changes introduced by ME-LoRA (i.e., what is currently Section 3.1).

- For the citations, please take a look at the ICLR template; citations should be in paranthesis, e.g.,
"the citation should be in parenthesis using \verb|\citep{}| (as in ``Deep learning shows promise to make progress
towards AI~\citep{Bengio+chapter2007}.'')."  However, this is not the case for most such citations in the paper, e.g.:
"the posterior distribution of the model parameters is inferred rather than relying on point estimates Bishop &
Nasrabadi (2006); Wang & Yeung (2020)"

- Please define the vec(\cdot) operator on line 181.  Also on line 181, \Sigma | A| should again be A^T \Sigma A.

- "Direct computation of the KL Divergence between the prior and posterior distributions of W is nontrivial. Direct computation of the KL Divergence between the prior and posterior distributions of W
is non-trivial."

- "3.2 EFFICIENT COMPUTATION OF FULL-WEIGHT KL DIVERGENCE" <- This is Theorem 3.2 from the BLOB paper (the title is exactly the same as the title used therein).

- Lines 215-230: "we adopt a strategy analogous to BLoB, where" <- This is called the reparameterization trick

- "However, with our proposed method, using such a simplistic prior variance can still lead to overfitting." <- Why?  Are there experiments to demonstrate this?  From the theoretical design of the KL-divergence in BLOB, it also hard to justify directly adding noise to the standard deviation \sigma_p (how could this arise given the Gaussian prior set up of the KL-divergence?).

---

### Note · Authors · 2024-11-21

I have read and agree with the venue's withdrawal policy on behalf of myself and my co-authors.